# Primary Metabolite Screening Shows Significant Differences between Embryogenic and Non-Embryogenic Callus of Tamarillo (*Solanum betaceum* Cav.)

**DOI:** 10.3390/plants12152869

**Published:** 2023-08-04

**Authors:** André Caeiro, Ivana Jarak, Sandra Correia, Jorge Canhoto, Rui Carvalho

**Affiliations:** 1Centre for Functional Ecology, Laboratory Associate TERRA, Department of Life Sciences, University of Coimbra, 3000-456 Coimbra, Portugal; andrecaeiro91@gmail.com (A.C.); sandraimc@ci.uc.pt (S.C.); 2Laboratory of Drug Development and Technologies, Faculty of Pharmacy, University of Coimbra, 3000-548 Coimbra, Portugal; jarak.ivana@gmail.com; 3i3S-Instituto de Investigação e Inovação em Saúde, Universidade do Porto, Rua Alfredo, Allen 208, 4200-393 Porto, Portugal; 4InnovPlanProtect CoLab, 7350-478 Elvas, Portugal; 5Department of Life Sciences, University of Coimbra, 3000-456 Coimbra, Portugal; isotopomero@gmail.com; 6REQUIMTE/LAQV, Group of Pharmaceutical Technology, Faculty of Pharmacy, University of Coimbra, 3000-456 Coimbra, Portugal

**Keywords:** chemometrics, differential metabolites, indirect embryogenesis, nuclear magnetic resonance spectroscopy, primary metabolism

## Abstract

Tamarillo is a solanaceous tree that has been extensively studied in terms of in vitro clonal propagation, namely somatic embryogenesis. In this work, a protocol of indirect somatic embryogenesis was applied to obtain embryogenic and non-embryogenic callus from leaf segments. Nuclear magnetic resonance spectroscopy was used to analyze the primary metabolome of these distinct *calli* to elucidate possible differentiation mechanisms from the common genetic background *callus*. Standard multivariate analysis methods were then applied, and were complemented by univariate statistical methods to identify differentially expressed primary metabolites and related metabolic pathways. The results showed carbohydrate and lipid metabolism to be the most relevant in all the *calli* assayed, with most discriminant metabolites being fructose, glucose and to a lesser extent choline. The glycolytic rate was higher in embryogenic *calli*, which shows, overall, a higher rate of sugar catabolism and a different profile of phospholipids with a choline/ethanolamine analysis. In general, our results show that a distinct primary metabolome between embryogenic and non-embryogenic *calli* occurs and that intracellular levels of fructose and sucrose and the glucose to sucrose ratio seem to be good candidates as biochemical biomarkers of embryogenic competence.

## 1. Introduction

Tamarillo, *Solanum betaceum* Cav. (=*Cyphomandra betacea* (Cav.) Sendt.), is a small solanaceous tree or shrub endogenous to South America [1]. Because of its edible fruits, it is currently grown globally, namely in New Zealand, Australia and California, USA [2]. Tamarillo fruits present a low caloric content and high concentration of vitamins and antioxidants, being consumed fresh or processed into several products such as jams or fruit-based juices [3]. The agronomical interest in tamarillo, caused by fruit market demand, lead to the necessity of efficient and reliable propagation techniques [4]. Furthermore, in this species, conventional propagation methods using seeds or grafting have proven to be insufficient due to several problems, namely the genetic variability of seeds or phytosanitary concerns [4,5]. Therefore, biotechnological cloning tools are a reliable alternative for selective tamarillo breeding and improvement. Several techniques involving the micropropagation of axillary shoots, organogenesis or somatic embryogenesis have been proposed [5]. Somatic embryogenesis (SE), in particular, since its first description in tamarillo in 1988, has been extensively studied and optimized [6,7]. 

SE is a complex process in which a somatic cell or tissue, under certain stimuli and without fecundation, originates a structure similar to a zygotic embryo [8]. This structure (a somatic embryo), after a series of developmental stages analogous to zygotic embryos, germinates into a plant [9,10]. Although this developmental pathway may occur naturally, it is overwhelmingly applied as an in vitro technology, with the aim of cloning specified genotypes or as a tool in plant developmental biology [11]. In tamarillo, as with other species, several distinct protocols for in vitro SE have been described using different explants (such as leaf segments from in vitro-grown clones and mature zygotic embryos) in varying growth conditions, like the types of plant growth regulators (PGRs) used [5,7]. In general, somatic embryogenesis is called indirect if there is the development of undifferentiated *callus* tissue before somatic embryo conversion, or direct if the presence of *callus* is minimal or absent [12]. Although SE is a widely employed technique with several technological applications and scientific potential, most in vitro protocols are achieved using trial and error, with molecular mechanisms and markers being poorly known in general [13,14].

In tamarillo, when the initial explants are leaf segments cultured in a high-sucrose culture medium supplemented with the synthetic auxin picloram, an indirect SE process occurs with *callus* formation after 12 to 14 weeks [7]. The *callus* tissue obtained can be maintained and multiplied in the same type of culture medium or converted into somatic embryos by stopping external auxin signaling and lowering sucrose concentration. In this protocol, it is also possible to distinguish two types of *callus* tissue according to their embryogenic competence: embryogenic *callus* (EC), which develops somatic embryos, and non-embryogenic *callus* (NEC) that is unable to evolve into somatic embryos and that becomes necrotic if picloram supplementation is cut [15]. Furthermore, it has also been shown that although both cell lines can be maintained indefinitely, embryogenic competence is somewhat time-dependent, as long-term *callus* lines begin to show genetic abnormalities and lower rates of somatic embryo conversion after being maintained in a culture for periods of more than 2 years [15,16]. The different phenotypes that arise from explants of the same genotype constitute an important feature of this system that has been optimized in several areas of application, namely genetics and epigenetics, and explored in several studies where the SE process was analyzed with complementary omics, namely proteomics [17,18]. 

Metabolite or metabolome-wide approaches have also been employed in plant model species to study several phenomena from the metabolic changes that occur in pollen germination [19] to long-term resistance phenotypes against biotic stresses in Arabidopsis thaliana [20]. In terms of in vitro biotechnological processes, metabolomic techniques have been employed in the study of the regeneration of *Solanum lycopersicum* [21] as well as SE studies in several important species such as *Coffea* sp. [14] and *Zea mays* [22]. These types of studies can be focused on particular sets of metabolites or metabolic pathways or use a wider approach by employing analytic techniques such as mass spectrometry or nuclear magnetic resonance (NMR) associated with chemometric methods to access the main metabolites of interest or to infer metabolic pathways involved in the responses [23,24]. Some works have also employed these methodologies in the search for biomarkers of a specific process, namely in the root development of *Daucus carota*, to assess the optimal harvest time and maximize bioactive compound accumulation in the final product [25]. In general, NMR protocols can be applied to several plant systems to acquire data that allow the application of the modern chemometric methods [23].

The objective of this study was the NMR-based metabolic fingerprinting of embryogenic and non-embryogenic *calli* obtained from an indirect SE protocol to identify the main metabolites present in each distinct phenotype, in order to evaluate the most important metabolites and metabolic pathways for embryogenic competence. This approach was combined with a standard chemometric analysis to derive models of metabolite relevance as well as a one-to-one differential analysis of different phenotypes using a long-term *callus* (LTC) as a reference state to help identify possible markers of embryogenic competence. The results are aimed at differentiating the two types of phenotypes in terms of the main metabolic pathways expressed and help to putatively identify possible biomarkers of SE in general or, more specifically, embryogenic competence.

## 2. Results

### 2.1. Callus Culture, Embryo Conversion and Protein Quantification

The induction protocol previously described led to the development of callus tissue on the original leaf explant. After approximately 12 weeks, several callus could be seed and morphologically distinguishable as embryogenic or non-embryogenic and were sub-cultivated as described in the Materials and Methods section. Additionally, a callus line that had been previously induced as embryogenic (EC), but lost that competence through subcultures, was used as long-term callus (LTC, Figure 1A). To confirm embryogenic competence, the calli were transferred to an auxin-free medium, and the mass increment (Figure 1B) and number of somatic embryos per gram of final mass were counted (Figure 1C). The EC lines presented an increment in mass during this period and somatic embryo conversion while NEC lines became necrotic, and no somatic embryos were observed. LTC callus showed little or no formation of somatic embryos. 

Although no statistical significance using an unpaired *t*-test (*p* < 0.05) was found between either the mass increment or the number of somatic embryos of embryogenic calli, EC2 presented slightly higher values of both (67.77 ± 1.87% and 17.67 ± 1.20 somatic embryos/gram, respectively).

Protein quantification was carried out on all the calli used in this work (Figure 2A). LTC showed the highest protein concentration (0.88 ± 0.08 µg of protein/mg of fresh weight) while NEC2 presented the lowest amount (0.13 ± 0.02 µg of protein/mg of fresh weight). Both embryogenic lines showed no statistical difference to either NEC1 or NEC2. NEC1 and NEC3 are also not statistically different when compared to NEC2. The dry-to-fresh mass ratio was also measured (Figure 2B), with EC2 showing the highest value (0.19 ± 0.03) followed by LTC (0.16 ± 0.02) and EC1 (0.14 ± 0.01), respectively, with statistically significant differences between EC1 and EC2. Compared with these lines, the three non-embryogenic calli presented a group with a significantly lower ratio (the lowest was NEC1 with 0.10 ± 0.002).

### 2.2. Metabolite Identification

Metabolome compositions of different cell lines were assessed with 1H-NMR spectroscopy. Qualitative spectral differences were apparent upon visual inspection (Figure 3A,B). Although a high number of metabolites was observed (>50), the spectral assignment using comparison with available public data bases and available literature led to the identification of 27 metabolites (Table 1). Some other metabolites, like those observed in the sugar region (5.15–5.6 ppm) or Phe congeners (7.1–7.5 ppm), still remain to be identified. Finally, due to extensive overlap or low representation, reliable peak integration was possible for 21 identified metabolites and they were included in a chemometric analysis (Table 1).

In the region from 0.6 to 4.5 ppm (Figure 3A), several proteinogenic amino acids (Ala, Asn, Asp, Gln, Glu, Ile, Pro and Val) were identified as well as the non-proteinogenic γ-aminobutyric acid (GABA). In terms of sugars, sucrose, fructose and glucose (α and β) were also found in all the calli lines. Other small intermediate metabolites (acetate, lactate and 4-hydroxybutyrate) were also identified in all samples. Choline was found in all the samples, as well as its possible precursor ethanolamine. Additionally, in this spectral region, two multiplet signals at 2.03 and 2.17 ppm were assigned to chlorogenic acid in EC1 and LTC.

In the low field region (5.2 to 9.4 ppm, Figure 3B), several aromatic metabolites were identified in all cell lines, and include amino acids phenylalanine (Phe) and tyrosine (Tyr). Formate, a monocarboxylic acid, was also found in all cell lines. Adenosine was identified in EC1, EC2 and NEC2. In terms of pyrimidines, uridine was found in the embryogenic calli (EC1 and EC2) and NEC2, whereas thymidine was only associated with NEC2. Nicotinamide was found in the non-embryogenic calli (NEC1, NEC2 and NEC3) and in EC1.

### 2.3. Chemometric Analysis

Although 21 identified metabolites represent only a fragment of the total *calli* metabolome, they contain sufficient information to distinguish between the tested cell lines. The ability of unsupervised PCA to preserve data variance and reduce dimensionality revealed the existence of distinct metabolite profiles of different *calli* since complete separation was observed in a scatter score plot (Figure 4).

The two principal components (PCs) account for 91.7% of the accumulated variance (Figure 4A), with all groups presenting good clustering without apparent outliers, although EC1, EC2 and LTC appear to show less intergroup variation. Fructose and, to a lesser extent, glucose appear to be highly related to PC1 and seem to be accumulated in NEC3, while sucrose and choline appear to be more connected to PC2 and contribute differentially to EC1 and LTC, respectively (Figure 4B).

Similar results were obtained with another unsupervised method. A hierarchical clustering dendrogram shows that each *callus* line forms a cluster, supporting the PCA analysis previously presented (Figure 5). EC2 is closer to NEC3, whereas EC1 appears to be closer to LTC. In this clustering analysis, the NEC3 *callus* appears to be the most dissimilar.

After the global analysis, pair-wise comparisons were analyzed to obtain more insight into the metabolome differences within the same type of *callus*. Therefore, EC1 vs. EC2 and NEC1 vs. NEC2 were built based on the proximity of the *calli* in the general model. Additionally, models comparing embryogenic lines and LTC and a model comparing the non-embryogenic *calli* provided insight into specific markers related to embryogenic competence and subculture stability. Performance parameters and important features for each model are summarized in Table 2.

After the multivariate analysis, a complementary investigation of differential expressed metabolites in the *calli* was performed with a univariate analysis. A volcano plot analysis revealed some common features between individual embryogenic and non-embryogenic cell lines when compared with the control LTC. In EC, four metabolites (lactate, glucose, proline and uridine) were commonly over-expressed and two (chlorogenic acid and tyrosine) were under-expressed. On the contrary, NEC cells presented a distinctly different common metabolic profile when compared with LTC with five metabolites (fructose, GABA, glucose and proline) being over-expressed in all *calli* and lactate being under-expressed. Additionally, each individual cell line displayed characteristic distinct metabolic differences. The fold changes of each metabolite were used for construction, as described in the Materials and Methods section.

The full list of metabolites in each group along with the respective value of log2(F) and −log10(*p*) are presented in Table 3.

### 2.4. One-Way Analysis and Metabolite Ratios

The metabolites were also analyzed with standard one-way statistical methods and the results are qualitatively shown in a cluster heatmap (Figure 6). All metabolites show some statistically significant differences across the five groups tested.

In general, the results support the previous findings, showing a large intergroup variation in sucrose, glucose and fructose with fructose showing the highest apparent concentrations. In terms of non-aromatic amino acids, some differences in the distribution pattern are also visible, with NEC2 usually presenting the highest values, without a clear clustering by phenotype. Tyrosine, one of the aromatic amino acids in this analysis, showed a higher concentration on LTC, with non-embryogenic *calli* presenting an intermediate group and embryogenic *calli* showing the lowest values.

Of the other metabolites, ethanolamine showed the highest value in NEC2 and the lowest in NEC1 with the remaining *calli* presenting intermediate values. In the lactate analysis, the embryogenic *calli* showed the highest values, followed by NEC2. NEC1 and NEC3 form an intermediate group and LTC shows the lowest value. Finally, choline concentration is statistically different in all *calli*, with LTC presenting the highest value and NEC1 the lowest.

For further information about the dynamic metabolic state of the *calli*, ratios of metabolites were calculated (Figure 7). The ratios of fructose to sucrose (Figure 7A) and glucose to sucrose (Figure 7B) presented similar patterns in the statistical analysis: the NEC *calli*, particularly NEC2 and NEC3, showed the highest values while NEC1 and EC2 presented an intermediate group and EC1 and LTC had the lowest values. The absolute value of the ratios, however, varies significantly in both types, being approximately ten-fold higher in the fructose/sucrose case.

In terms of the glucose to lactate ratio (Figure 7C), NEC3 presents the highest value followed by NEC1. The remaining *calli* did not show statistically significant differences. The choline/ethanolamine ratio (Figure 7D) appears as higher in LTC followed by EC1 and EC2 while the non-embryogenic presents lower values with NEC3 and NEC2 being the lowest.

## 3. Discussion

### 3.1. Somatic Embryogenesis and Metabolite Profile

The protocol used in this work can be defined as an indirect two-step process, in which dedifferentiated *callus* tissue develops from the initial explant on an auxin-rich medium, but somatic embryo conversion, from induced embryogenic cells, is only achieved if the external auxin stimulation is interrupted [5]. The emergence of two distinct phenotypes of *callus*, distinguishable by embryogenic competence, is observed in several protocols initiated from distinct explants, namely zygotic embryos or, as in the present case, young leaf segments [16], with previous works already exploring differences between EC and NEC *calli* in tamarillo [26]. Similar SE protocols have been described in other species such as Arabidopsis thaliana and cotton (*Gossypium hirsutum*), with studies also employing embryogenic and non-embryogenic *callus* to determine the molecular mechanisms underlying the emergence of both phenotypes and understand the SE process as a whole [27,28]. Furthermore, EC was also used in genetic modification protocols [29].

The EC derived from this work shows a similar (statistically insignificant) embryogenic competence but varies in terms of both total protein content and fresh-to-dry mass ratios, revealing some heterogeneity in the first analysis. However, differences in the somatic embryo conversion of several *calli* lines have been previously described in tamarillo [26], as well as other species, namely Norway spruce (*Picea abies*), in which the conversion of somatic embryos from embryogenic *callus* is considered by the authors as one of the most important factors for the large-scale implementation of SE protocols [30]. The NEC *calli* show a greater similarity in the mass ratios but a higher disparity in total protein content. These observations are in accordance with previous studies of these types of *calli* that have shown EC and NEC *callus* to be morphologically distinct [31] and present the NEC *calli* as a highly hydrated tissue. Furthermore, previous studies have shown some differences at the proteomic level on similar *callus* lines [26] and quantitative and qualitative differences in EC- and NEC-*callus*-secreted proteins [32].

### 3.2. Identified Metabolites and Possible Biosynthesis Pathways

The main metabolites identified in this work are consistent with those described in the literature for *callus* obtained in SE studies. Free amino acids, for example, have been found in cultures of *Coffea arabica* L. [33]. Other studies have also examined the importance of these metabolites with proline, being related to SE regulation in suspension cultures of *Dactylis glomerata* [34] and stimulation in *Cuminum cyminum* [35]. Acetate, ethanolamine, *calli* choline and formate have been described in *Saccharum* sp. suspensions of *calli* derived with SE protocols in an NMR-based approach [36]. Chlorogenic acid and some derivatives have also been found and closely linked to SE in *Coffea arabica* [14].

The *calli* used in this study were induced and maintained in a Murashige and Skoog medium supplemented with sucrose and picloram. Consequently, glucose and fructose found in *calli* can be inferred to have derived from sucrose. However, the culture medium contains only the basic inorganic salts needed for plant growth and some organic compounds: myo-inositol, nicotinic acid, pyridoxine and thiamine. Furthermore, the *calli* underwent two or three cycles of in vitro multiplication to achieve enough mass for the subsequent tests (chemometrics and conversion assays). Therefore, the metabolites detected were synthetized by the *calli* in anabolic pathways that start in intermediates derived from the catabolic activity. Lactate is an end-product of glycolysis [37]; formate can have several origins in plants. In photosynthetic tissues, it can be linked to photorespiration, while in non-photosynthetic tissue, it can arise as a fermentation end-product [38]. Acetate is a key metabolite in the interface between carbohydrate and lipid metabolism and can be obtained from acetaldehyde, a product of either the decarboxylation of pyruvate or the dehydrogenation of ethanol [39].

Amino acid synthesis in plants is highly complex and is connected with several metabolic pathways, notably carbohydrate metabolism and namely the tricarboxylic acid (TCA) cycle, where 2-oxoglutarate is an important metabolite that can originate the non-aromatic proteinogenic amino acids [40,41]. Other compounds also derive directly or indirectly from amino acids: GABA can be derived from glutamate [42] and 4-hydroxybutyrate is involved in GABA metabolism, namely in the NADPH-assisted reduction of succinic semialdehyde, an extremely reactive intermediate of the GABA pathway [43,44]. Ethanolamine can be obtained with the decarboxylation of serine [45] and is subsequently used to synthetize choline [46], with both molecules being key compounds in the Kennedy pathway that originates the main cellular phospholipids in eukaryotes [47].

Additionally, the identified aromatic amino acids (tyrosine and phenylalanine) and chlorogenate are products of the phenylpropanoid pathway [48,49] that primarily originates in the condensation of phosphoenolpyruvate (glycolytic intermediate) and erythrose 4-phosphate, an intermediate of the pentose phosphate pathway [50]. Lastly, the de novo synthesis of nucleotide (and nucleosides) can occur from precursors found in the *calli*, namely the amino acids that can be derived from the main carbohydrate metabolism, particularly the TCA cycle. Additionally, nicotinic acid as a nucleotide precursor has been extremely well reviewed [51,52].

In general, the described metabolic reactions have not been specifically identified in the tamarillo plant or *callus* tissue; however, the overall pathways are similar in many plant families. For example, the Shikimate (phenylpropanoid) pathway is common to several plant families [53] and, although some of the detected molecules may have transitioned from the metabolome of the original explant (leaf segments), with the multiplication and the composition of the culture medium, the presented synthetic pathways are very likely to account for the metabolites detected.

### 3.3. Chemometric Analysis

Following spectral processing and the metabolite identification and construction of the proper multiplet-based data matrices, multivariate statistical methods were applied to the resulting data, namely a principal component analysis (PCA) followed by a Projection to Latent Structures Discriminant Analysis (PLS-DA). Similar methodology workflows have been described and are often employed to address the type of data that results from a metabolic screening based on NMR spectroscopy [54,55]. PCA is typically the first method used as it allows for visualizing, with some limitations, the broad structure of the data in terms of the most important metabolites and inferring some important tendencies, such as a putative identification of potential biomarkers [56]. PCA is an unsupervised method that reduces the dimensionality of the data set while maximizing the associated variance by extracting orthogonal elements called principal components (PC) using the correlation matrix of the data [57]. The results found here show a good separation of all the *calli* samples along the first two PCs with fructose, glucose, sucrose and choline being the greatest contributors to the overall variance. The medium sugar concentration has long been recognized as an important factor in increasing SE efficiency in several species, namely in *Cucumis sativus* [58] and tamarillo [16,26], with several authors associating the increase in efficiency largely (but exclusively) with osmotic effects [59,60]. More recently, in metabolomic-orientated studies in coffee, glucose and fructose have been found as over-accumulated in embryogenic *calli* [14]. Altogether, these results support the findings of this work, and suggest that although osmotic pressure effects might be relevant for SE, carbohydrate-related pathways are extremely important for the SE process. This is in accordance with the differential expression of several metabolism-related proteins previously identified when comparing tamarillo ECs and NECs [26]. Other studies have shown the importance of lipid content in SE in *Cichorium* sp. and *Picea asperata* [61,62], and in tamarillo somatic embryos [18], which could explain the importance of choline in the overall variance of the data.

The dendrogram presented was obtained with a hierarchical clustering analysis, a type of classification procedure where a specific distance metric with an associated classification scheme is applied to the data, leading to the classification of the instances based on shared similarity [63]. Such methodologies have been employed in several metabolite-based studies [22,64] to classify different types of phenotypes in terms of a certain response or to study specific phenomena. The results here show a clear grouping of samples with all clusters being formed by *calli* from similar groups; however, the relative distance between the groups does not follow an immediate pattern (EC or NEC clusters close to each other), hitting the high heterogeneity of the cell populations in the study. Previous studies in other species such as *Olea europaea* and *Arabidopsis thaliana* have shown a great diversity of the *callus* response in terms of in vitro protocols [27,65]. The data presented here show this type of heterogeneity in the cell lines studied.

The next step of the analysis used a supervised method, namely a Projection to Latent Structures Discriminant Analysis (PLS-DA), a technique that is amply used in chemometric studies [66]. PLS-DA is a generalization of the underlying principles of PCA and multivariate regression where dependent variables are extracted from the data, by finding a set of orthogonal factors (latent variables) that maximize predictive power, which is tested with specific methodologies such as cross-validation schemes [67]. Such methodologies have been employed in studies of somatic embryogenesis to evaluate proteomic or metabolic data [14,68]. Thus, the objective of the application of this method in the present context was to evaluate the most discriminant metabolites in terms of the different phenotypes tested, which justifies the several models built. Fructose and glucose are the only metabolites that appear in all the models, being, in line with the previous PCA analysis conducted, the most important metabolites in terms of the overall structure of the data. In general terms, the PLS-DA analysis shows that sugar metabolism is the main metabolic process in the discrimination of the several *calli* studied, with lipid metabolism, represented in some models by choline, also being relevant (in fact, choline only presents a VIP lower than the threshold of 1 in the models of embryogenic *callus* discrimination—EC1 vs. EC2). The amino acid proline, over-expressed in some EC and NEC, has been linked to SE as a stimulant agent [69] and indicated, in general, as involved in stress regulation mechanisms [70]. Interestingly, this metabolite only appears as relevant in the models that present two groups that are closely grouped (EC1 to EC2 and NEC1 to NEC2), which could hint that proline or, more generally, the intracellular stress level or stress-regulating mechanisms are important to distinguish between metabolic identical *calli* with the same embryogenic phenotype. In fact, some studies have already highlighted different stress response mechanisms in EC and NEC in *Vitis vinifera* [71] as well as in tamarilho [72]. Further studies should address this question [71].

After the analysis of the general data structure and the main metabolites involved in *calli* separation, a volcano plot analysis was used to address differentially expressed metabolites in the embryogenic and non-embryogenic *calli* using LTC as a common reference. This type of univariate analysis can be considered complementary to the multivariate analysis used [73] and is frequently applied in data derived from metabolic/metabolome studies [74], combining a fold analysis (average concentration of a given metabolite in terms of the control) with a parametric or non-parametric statistical test, allowing the visualization and screening of the most important factors in the direct comparison between two groups, presenting some limitation if sample outliers are present [21,75]. The results show that the profile of over/under-expressed metabolites is more uniform in embryogenic *calli* when compared to the non-embryogenic groups, supporting the global statistical analysis that shows NEC *callus* as having a greater intrinsic heterogeneity. Sucrose and its immediate hydrolysis products appear in several analyses as either over-expressed or under-expressed, whereas choline and chlorogenate are more uniformly over-expressed in LTC. In general, these results appear to support the hypothesis that the carbohydrate is one of the main determinant factors to the embryogenic phenotype, as found in this work and previously described [76,77], while some lipid-related metabolism as well as some secondary anabolic pathways are specific to LTC, or at least some precursors are notably present. The subsequent step in the analysis, focusing on some of the key metabolites described, aimed to increase the insight into the overall metabolic state of each cell group and try to find the most viable candidates as biomarkers of embryogenic competence in tamarillo *calli*.

### 3.4. One-Way Analysis and Metabolite Ratios

Sucrose is the main sugar transported in plants and, as such, its homeostasis processes have been extensively studied. In cellular uptake, the involved protein transporters are largely, but not exclusively, sucrose/H+ symporters [78,79]. In an intracellular medium, sucrose can be hydrolyzed by the action of either invertase (EC 3.2.1.26) into glucose and fructose, or sucrose synthase (EC 2.4.1.13) into UDP-glucose and fructose [80], with both types of enzymes having been identified in members of the Solanaceae family, namely *S. lycopersicum*, and studied in the context of several physiological phenomena [81,82]. Of these three metabolites, only sucrose was in the initial medium formulation; therefore, it can be concluded that the intracellular import of this metabolite is active in all the *calli* and assumed that at least a significant fraction of glucose and fructose are products of sucrose hydrolysis. In fact, the higher ratios of both glucose and fructose to sucrose found in all the *calli* may support this hypothesis by suggesting a high activity of hydrolysis. Previous studies using stable isotope labelling on tomato suspensions have shown no gluconeogenesis [45], which further supports this assumption, whereas the different concentrations of intracellular sucrose may point to distinct transport kinetics, although further studies would be needed to study the sucrose import phenomena, namely in terms of the identification and characterization of the specific transporters used. Furthermore, the concentrations of glucose and fructose are higher in NEC *calli* when compared to EC (NEC1 is not statistically significant), which, assuming the high hydrolysis and low biosynthesis rate hypothesis, might indicate an overall lower metabolic activity of NEC in terms of carbohydrate metabolism. In fact, previous proteomic-based studies in similar *calli* have shown different levels of the expression of several glycolytic and TCA enzymes, with some key enzymes (enolase and fumarate hydratase 1) being over-expressed in EC *callus* [72]. This observation can be further investigated by analyzing the glucose-to-lactate ratio, which has been routinely used to determine glycolytic rates in NMR-based studies [83,84]: EC1 and EC2 tend to have the lowest values, indicating a higher glycolytic rate, meaning that a larger fraction of the intracellular glucose is being actively employed in catabolism. Future studies should aim for a deeper analysis of this part of the main energetic pathways employed by each *callus* to elucidate, for example, what is the relation between glycolytic and oxidative catabolism.

In terms of the amino acid analysis, proline shows a greater variance between groups of phenotypes than all other metabolites individually analyzed. As previously stated, this amino acid has been closely linked to the stress response. Therefore, it could be linked to individual stress mechanisms of each *callus* line and not very useful as a discriminant metabolite, although it could be employed to indicate different stress levels within the same phenotype. Alanine, although not as connected with stress response phenotypes as proline, is an important amino acid in several physiological responses and may be related to the overall energetic availability of the cell [85]. This metabolite can be obtained from the reversible reaction catalyzed by alanine aminotransferase (EC 2.6.1.2), a ubiquitous enzyme in plants [86]. In general, alanine presented high concentrations in embryogenic *calli* and lower values in LTC and non-embryogenic *callus* (with the exception of NEC2), which might indicate some importance in embryogenic competence and an implicit regulation mechanism of the concentration of this metabolite. Tyrosine presents a distinct profile from all other amino acids found, with the lowest levels in EC *calli*. This pattern can be interpreted as a result of the higher rate of protein synthesis in these *calli* or possibly in terms of the underlying pathway of synthesis. Furthermore, as previously described, the aromatic amino acids are synthetized in the Shikimate pathway, an important anabolic route of the synthesis of several classes of secondary metabolites and regulatory factors [87]. Therefore, these results might suggest that tyrosine in non-embryogenic *calli* and in LTC (in this case with chlorogenic acid) represents an end-product of the Shikimate pathway: either by not being incorporated into proteins or by the lack of metabolic branching originating other compounds. GABA is a non-proteinogenic amino acid that has been recently related to several phenomena such as SE, by influencing several important pathways, namely the auxin-related regulation routes [88,89]. In the *calli* studied, GABA does not appear to be closely linked to embryogenic competence in general but to specific non-embryogenic phenotypes.

Finally, choline and ethanolamine and their ratios provide indirect information about the phospholipid profile of the *calli* lines. As previously discussed, both molecules are key metabolites in the biosynthesis of the main phospholipids found in eukaryotes, namely phosphatidylcholine and phosphatidylethanolamine, with ethanolamine being a precursor to choline [47]. Therefore, the concentrations of each metabolite could indicate the types of phospholipids preferred, and the ratio of the two can be proportional to the biosynthesis of choline, with greater ratios representing a larger fraction of ethanolamine converted to choline. The total concentration of each metabolite does not present a clear separation of groups. However, upon calculating the ratio, the separation of the NEC and EC *callus* is achieved: although there are still statistically significant differences between EC1 and EC2, the embryogenic *callus* shows a higher value of this ratio. Taking the assumptions previously described, this result might indicate higher concentrations of phosphatidylcholine. As previously described, both metabolites originate in serine, which can be synthesized from the glycolytic intermediate 3-phosphoglycerate [90], again reinforcing the central role of the carbohydrate metabolism in these *calli* and, consequently, in embryogenic competence.

The development of complementary omics (genomics, transcriptomics, proteomics and epigenomics) in the past two decades led to an increased understanding of the flow of information encoded by the genome, and provided insight into molecular processes in living organisms. At the top of the omics pyramid, the study of the metabolome reflects the information expressed and modulated by other omics. As the closest reflection of the phenotype, metabolomics is a rich source of biomarkers but also provides information on biological roles of metabolites as well as their interaction with the other omics. As such, it is an integral part of basic and applied studies of plant development and function [91]. Unlike other omics, the metabolome is highly sensitive to stimuli and provides a snapshot of the real-time physiological state of the observed system. Additionally, complex mechanisms of information transfer between other omics layers (post-transcription and post-translation) do not always allow omics like transcriptomics and proteomics to correlate with the observed phenotype [92]. With this in mind, we performed an untargeted analysis of the *calli* metabolome to obtain insight into the principal metabolic mechanisms that drive somatic embryogenesis. The results of our study describe the response of various metabolic pathways during the SE, with the major contributions of carbohydrate metabolism, as was previously observed for other species [77]. Despite the intriguing findings of this study, the approach based on only one omics provides only a fragmentary picture of this complex process. Since metabolite content reflects a complex interaction of other omics levels under given conditions, to complete the understanding of the SE, the results of this study should be complemented by additional experiments. Although the relative insensitivity and low resolution of the NMR technique provide limited insight into the metabolome, the results of this study offer valuable information about the SE in this plant as well as guidelines for future research within complementary omics.

## 4. Materials and Methods

### 4.1. In Vitro Propagation and SE Induction

The *calli* tissue used in this work were obtained from leaves of previously established shoot cultures of tamarillo seedlings (red variety) germinated in vitro. After germination, plantlets were cloned through the culture of apices or nodal segments on a Murashige and Skoog (MS) medium [93] supplemented with 8.6 mM of sucrose, 0.88 µM of 6-benzylaminopurine (BAP) and 6 g/L of agar (Sigma-Aldrich, St. Louis, MI, USA) as a gelling agent. For further multiplication, nodal segments (1–1.5 cm) were taken monthly and subcultured in the same medium. Shoot cultures were kept in a growth chamber at 25 °C, in a 16 h photoperiod, at 25–35 μmol m^−2^s^−1^ (white cool fluorescent lamps).

The apical leaves from shoots were used for SE induction as previously described [7,16] with small modifications: leaves were removed from the plants, segmented into areas of approximately 0.2 cm^2^, randomly punctured on the abaxial side (3 to 4 punctures per segment) and placed on Petri dishes (9 cm in diameter and 15.9 mm in height) with 35 mL of the MS medium supplemented with 26 mM of sucrose, 20 µM of picloram and 2.5 g/L of PhytagelTM (Sigma-Aldrich, St. Louis, MI, USA) as a gelling agent. The explants were kept in the dark at 24 ± 1 °C for 12 to 14 weeks until *callus* development was apparent. At this point, *callus* samples from apparently embryogenic (EC1 and EC2) and non-embryogenic (NEC1, NEC2 and NEC3) phenotypes were selected and subcultured in the same medium for multiplication. Each subculture lasted 1 month and samples for a metabolic analysis were retrieved after 2 to 3 subcultures. Additionally, a long-term *callus* (LTC) previously established from the same clones and continuous subcultures (more than 3 years) that had shown a loss of embryogenic competence were also used.

To confirm the embryogenic competence of the *calli* lines, a mass of about 250 mg was transferred to Petri dishes with the MS medium supplemented with 11.6 mM of sucrose and 2.5 g/L of PhytagelTM as a gelling agent and kept in the dark at 24 ± 1 °C for 5 weeks. The final mass was weighed and the number of somatic embryos per mass was registered. The mass increment of the *callus* during the conversion period was registered as a percentage of the initial mass ((final mass − initial mass)/initial mass × 100). Conversion assays were carried out in triplicate.

All culture media used in this work were adjusted to pH 5.7 (using either HCl or KOH) before adding the gelling agent and autoclaved at 121 °C for 20 min.

### 4.2. Metabolite Extraction

The *calli* samples were collected and immediately ground in liquid nitrogen before extraction in a methanol–water solution with pH 7.0 (2 mL/g FW). Similar solvent systems have been previously described in the literature and have been shown to be effective in these cell lines [72]. Furthermore, the neutral pH facilitates the usage of pre-existing proton chemical shift databases [93]. After centrifugation (16,000× *g*, 10 min, 4 °C), the supernatants were evaporated at 65 °C (approximately 24 h), and the metabolites were resuspended in a phosphate-buffered (0.2 M) D2O solution before an NMR analysis.

### 4.3. Protein Quantification

For soluble protein quantification, samples were ground in liquid nitrogen and extracted (2 mL/FW) in a 0.1 M sodium phosphate buffer, pH = 7.0. The extract was centrifuged (16,000× *g*, 10 min, 4 °C) and the supernatant was used for quantification with a Bio-Rad Protein Assay based on Bradford’s reaction [94] in a 96-well microplate. Bovine serum albumin (BSA) was used to construct a calibration curve between 5 and 40 µg/mL and absorbance was read at 595 nm in a SPECTRAmax PLUS 384 spectrophotometer. All measurements (standards and samples) were made in triplicate. In all samples, a fresh amount of *callus* (75 mg) was immediately weighed and dried for 72 h to determine the dry to weight ratios. These determinations were also made in triplicate.

### 4.4. Nuclear Magnetic Resonance and Chemometric Analysis

The aqueous cell extracts were thawed, homogenized using a vortex and centrifuged (9200 rpm, 5 min). 1H NMR spectra of the extracts were acquired on a 500 MHz Brucker Advance III HD Spectrometer (11.7 T) equipped with a 5 mm double-resonance broadband probe, using the following acquisition parameters: 298 K, a 5.7 kHz spectral width, a 0.1 s mixing time, 2 dummy scans, a total relaxation time of 7 s, a 90 °C pulse angle, a 2.85 s acquisition time and 64 scans. Spectra were processed by multiplying FIDs with the exponential apodization function (line broadening of 0.3 Hz) prior to Fourier transformation. Additionally, spectra were manually phased in TopSpin 4.0.8 (Bruker Biospin, Karlsruhe, Germany). Further spectral processing and bucketing were performed on the online platform NMRProcFlow [95]. Chemical shifts were internally referenced to a fumarate singlet at 6.5 ppm. Baseline correction and spectral alignment were based on adaptive iteratively reweighted penalized-least-square and least-square algorithms, respectively. Variable bucketing containing metabolite multiplet areas was used to create data matrices for a multivariate analysis. Buckets were normalized using the total integral area to account for matrix dilution effects and experimental conditions [96,97]. Metabolites were identified according to Metabolomics Standards Initiative (MSI) guidelines [98]. Obtained normalized areas representative of metabolite concentrations were analyzed with a univariate and multivariate statistical analysis. The multivariate analysis was performed on the online platform MetaboAnalyst 5.0 (www.metaboanalyst.ca accessed on 29 June 2023). Variables (metabolites) were submitted to Pareto scaling prior to the multivariate analysis. A principal component analysis (PCA) evaluated the initial data structure and was used to identify outliers and evaluate possible clustering. A hierarchical clustering dendrogram (HCA) was also constructed with the Ward clustering algorithm using Euclidean distance. Next, a supervised method, a Projection to Latent Structures Discriminant Analysis (PLS-DA), was applied to identify differentially expressed metabolites. Metabolites with a PLS-DA VIP > 1 were considered relevant to group separation. PLS-DA models validated by 5-fold cross-validation and the R2Y and Q2 were used to assess the fitting validity and predictive abilities of the models. Metabolite assignment was based on reference spectra available in public databases such as HMBD. Differentially expressed metabolites were further analyzed with volcano plots using LTC as a reference state.

### 4.5. One-Way Analysis and Metabolite Ratios

After the multivariate analysis, a differential univariate analysis with volcano plots (*t*-test was used with a *p*-value of 0.01, FDR-corrected, and a fold change threshold of 2) with LTC as the reference state was applied to all samples. To help visualize the results, fold changes of the significant metabolites were calculated for each sample (the non-significant fold changes were set to 0) and a heatmap based on mean aggregation was built in MATLAB^®^. Furthermore, all the identified metabolites were analyzed with standard one-way statistical methods, namely an analysis of variance (ANOVA) followed by a Tukey test (*p* < 0.05). These results were presented using a heatmap based on the normalized data using Euclidean distance as the metric and Ward’s clustering method.

Finally, ratios of some metabolites (fructose/sucrose, glucose/sucrose, glucose/lactate and choline/ethanolamine) were also studied. These ratios were analyzed with standard one-way statistical methods. In general, the homogeneity of variances was tested with the Brown–Forsythe test (*p* < 0.05). If this test was passed, the analysis of variance (ANOVA) followed by a mean comparison with the Tukey test (*p* < 0.05) was used. If the homogeneity of variances was not met, a Kruskal–Wallis non-parametric ANOVA followed by Dunn’s multiple comparison test was used. These results were analyzed and plotted in GraphPad^®^ Prism v 8 for Windows.

## 5. Conclusions and Future Perspectives

Somatic embryogenesis is a complex process that has been extensively studied in several species, including tamarillo. In this work, metabolic-based data were obtained using NMR spectroscopy in embryogenic and non-embryogenic *calli*. The analysis shows several metabolites with a high explanatory power in terms of the overall system and in the classification of the two morphogenesis phenotypes observed as well as some import differentially expressed metabolites. In general, fructose and glucose, the direct products of sucrose hydrolysis, are fundamental. Choline and some amino acids are also important to the overall data acquired. The main pathways by which these metabolites could have arisen in this system and some physiological implications were discussed.

With the data acquired, information about the main pathways used in indirect SE was also drawn, with embryogenic *calli* presenting a higher level of glycolysis in particular and an overall higher inferred metabolic rate showing more sugar conversion. Phospholipid metabolism, represented by choline and ethanolamine, was also shown to be distinct in the two phenotypes studied.

The glucose or fructose to sucrose ratio, the glycolytic rate as measured with the glucose to lactate ratio, the choline concentration and choline/ethanolamine appear as good indicators of embryogenic competence. To a lesser extent, Shikimate-related metabolites might be linked to embryogenic *callus.*

Overall, the results found here are complementary to others described in the literature, particularly in terms of tamarillo proteomics, and show that the two different phenotypes previously described in terms of somatic competence (embryogenic and non-embryogenic) vary in terms of the proteome and metabolome. The data also suggest that the simple determination of specific metabolites (glucose, fructose and lactate) can be used to infer the embryogenic competence of *callus* lines, either upon the final induction protocol or the stability of this characteristic along time, which is a faster and easier methodology than undergoing the conversion process currently applied. In future works, the prediction power of these indicators should be further tested.

Future studies should aim to further characterize this *calli*, particularly in terms of specific metabolic fluxes of glycolysis and TCA, to achieve a higher degree of information about glycolic and oxidative catabolism as well as the connection with the anabolic pathways necessary to achieve the metabolites found in this and other studies.

## Figures and Tables

**Figure 1 plants-12-02869-f001:**
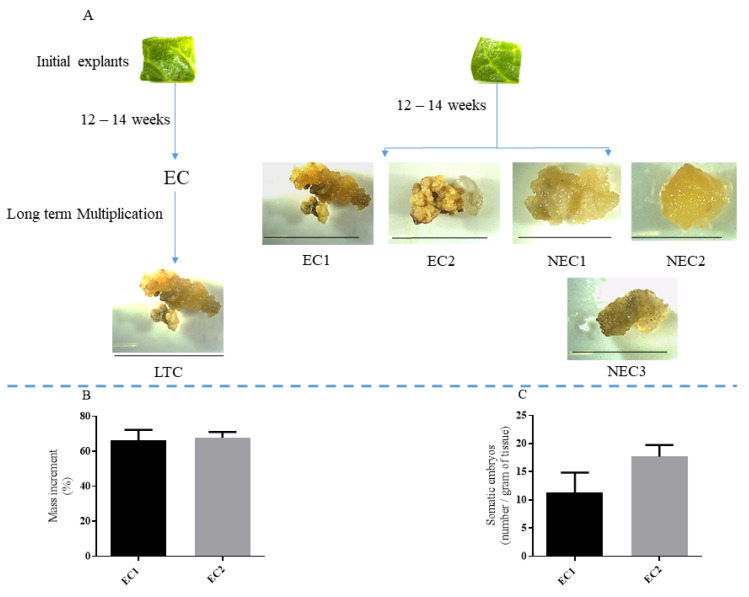
Calli sampled for metabolic analysis. (**A**) Induction steps used to obtain the calli used in this work. Bar represents 1 cm. (**B**) Mass increment of EC1 and EC2. (**C**) Number of somatic embryos observed after 5 weeks in conversion medium (MS supplemented with 11.6 mM of sucrose).

**Figure 2 plants-12-02869-f002:**
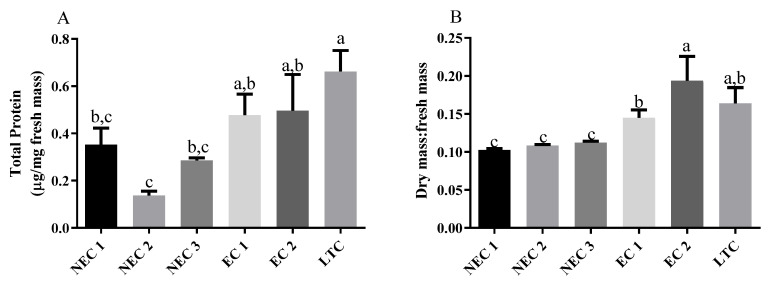
General *calli* parameters. (**A**) Protein quantification of the several *calli*. (**B**) Dry to wet mass ratio. Different letters are significantly different according to Tukey test (*p* < 0.05).

**Figure 3 plants-12-02869-f003:**
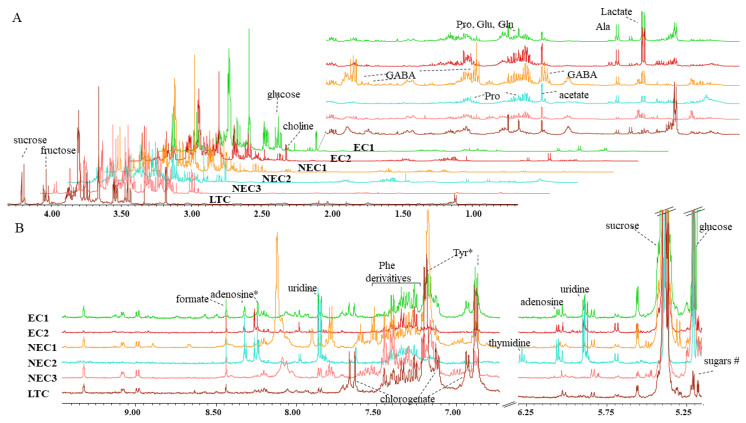
Average 1D ^1^HNOESY NMR spectra of *calli*. Spectral regions were segmented for better insight. (**A**) A total of 0.5 to 4.5 ppm with further augmented region from 0.6 to 3.0 ppm. (**B**) A total of 5.2–9.4 ppm where water and fumarate regions (4.3–5.2 ppm and 6.31–6.75 ppm, respectively) were excluded. Major identified metabolites were annotated in different spectral sections. Abbreviations: three-letter code for amino acids. * metabolites identified as members of a compound class based on spectral similarity to known compounds of a chemical class (level 3) according to the MSI guidelines; # unidentified sugars.

**Figure 4 plants-12-02869-f004:**
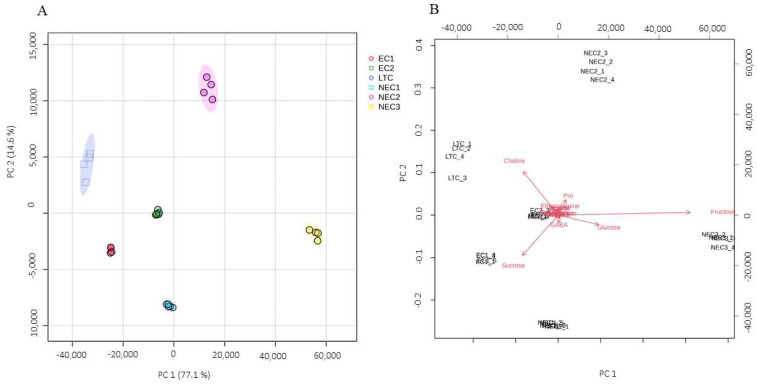
Principal component analysis of the six *calli* lines used. (**A**) Score scatter plot. Principal components PC1 and PC2 and related explained variability (%) are indicated on axes x and y, respectively; (**B**) biplot presents score scatter plot overlapped with variable loading vectors that denote variations in differentially expressed metabolites.

**Figure 5 plants-12-02869-f005:**
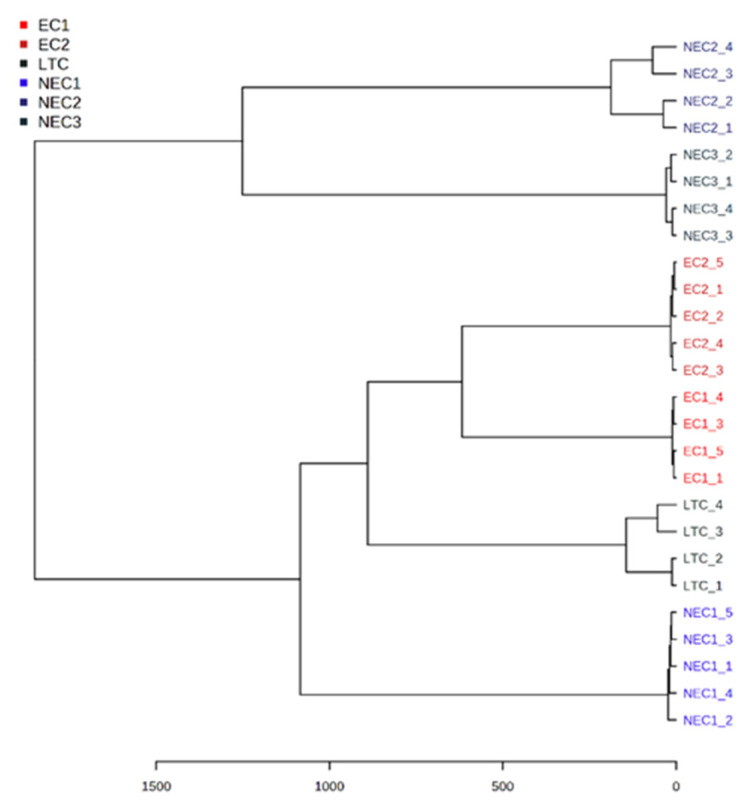
Dendrogram of hierarchical cluster analysis. Different *calli* are represented by different colors; the similarity levels are expressed in Euclidean distance.

**Figure 6 plants-12-02869-f006:**
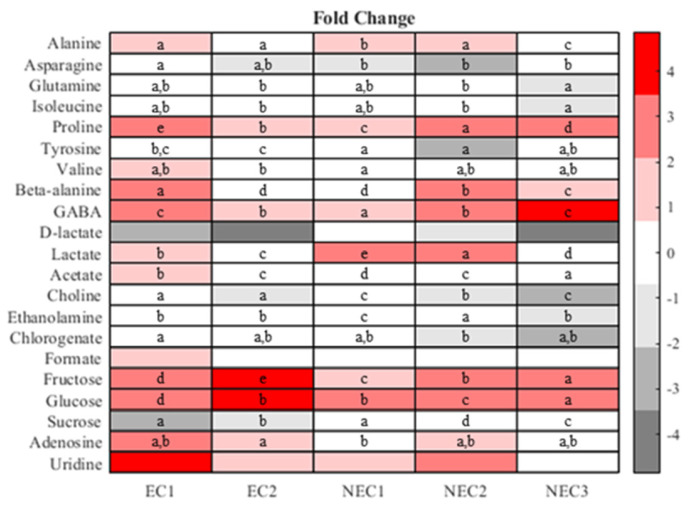
Heatmap of fold change. The fold change was calculated against the average concentration of each metabolite in the LTC *callus*. Different letters represent statistically different metabolites with *t*-test (*p* < 0.05).

**Figure 7 plants-12-02869-f007:**
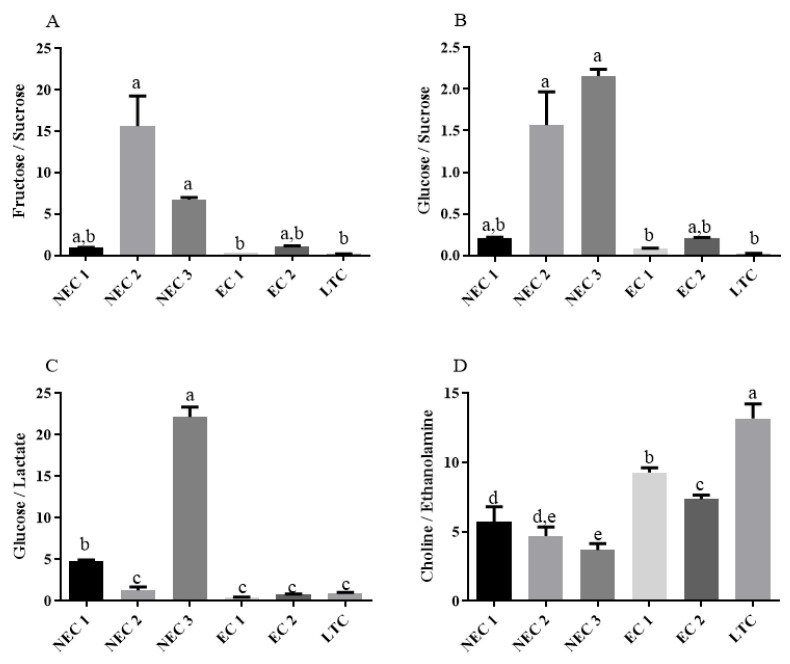
Metabolite ratios. (**A**) Fructose to sucrose. (**B**) Glucose to sucrose. (**C**) Glucose to lactate. (**D**) Choline to ethanolamine. Different letters are significantly different according to Tukey test (*p* < 0.05).

**Table 1 plants-12-02869-t001:** Assignment of resonances in the NMR spectra of the *calli*. (s: singlet, d: doublet, t: triplet, m: multiplet, dd: doublet of doublets).

	Compound ^1^	Assignment	δ ^1^H ppm (Multiplicity)	Cell Type
1	Acetate	*α*CH_3_	1.9 (s)	all
2	Adenosine/inosine	C1’H ribose	6.05 (d)	NEC2, EC1, EC2
		C8H ring	8.24 (s)	
		C2H ring	8.32 (s)	
3	Alanine	*β*CH_3_	1.46 (d)	all
4	Asparagine	*β*CH	2.84 (dd)	
		*β*’CH	2.94 (dd)	
5	Aspartate *	*β*CH	2.66 (dd)	all
		*β*’CH	2.79 (dd)	
6	*β*-alanine	*α*CH_2_	2.54 (t)	NEC1-3, EC1, EC2
		*β*CH_2_	3.14 (t)	
7	Choline	CH_3_	3.18 (s)	all
		CH_2_ (N)	3.49 (m)	
		CH_2_ (OH)	4.04 (m)	
8	Chlorogenic acid	C3H	2.03 (m)	EC1, LTC
		C5H	2.17 (m)	
		C16H	6.40 (d)	
		C21H	6.93 (d)	
		C22H	7.12 (dd)	
		C18H	7.18 (d)	
		C15H	7.65 (d)	
9	Ethanolamine	C2H_2_	3.12 (m)	all
		C3H_2_	3.81 (m)	
10	Fructose	C1H	3.55 (m)	all
		C1H, C6H	3.68 (m)	
		C1H, C2H, C5H	3.77 (m)	
		C4H	3.88 (m)	
		C5H	3.99 (m)	
		C6H	4.02 (m)	
		C3H, C4H	4.09 (m)	
11	Formate	CH	8.43 (s)	all
12	*γ*-aminobutyric acid (GABA)	C2H	1.88 (m)	all
		C3H	2.28 (t)	
		C4H	2.99 (dd)	
13	*α*-glucose	C4H	3.38	all
		C2H	3.53	
		C3H	3.76	
		C5H, C6H	3.82	
		C1H	5.21 (d)	
	*β*-glucose	C2H	3.22	
		C4H	3.38	
		C5H, C3H	3.45	
		C6H	3.72	
		C6’H	3.87	
		C1H	4.62 (d)	
14	Glutamate *	*β*CH	2.05 (m)	all
		*β*’CH	2.12 (m)	
		*γ*CH_2_	2.35 (m)	
15	Glutamine	*β*CH_2_	2.14 (m)	all
		*γ*CH_2_	2.45 (m)	
16	4-hydroxybutyrate *	C3H	1.8 (m)	all
		C2H	2.21 (dd)	
17	Isoleucine	*δ*CH_3_	0.94 (t)	all
		*β*’CH_3_	1.01 (d)	
18	Lactate	*β*CH_3_	1.31 (d)	all
		*α*CH	4.11 (q)	
19	D-lactate	*β*CH_3_	1.38 (d)	all
20	Nicotinamide ^2,^*		8.98 (d)	NEC1-3, EC1
			9.08 (d)	
			9.32 (s)	
21	Phenylalanine *	C2H, C6H ring	7.31 (m)	all
		C4H ring	7.36 (m)	
		C3H, C5H ring	7.40 (m)	
22	Proline	*γ*CH_2_	1.97 (m)	all
		*β*’CH	2.05 (m)	
		*β*CH	2.33 (m)	
		*δ*CH_2_	3.31 (m)	
		*α*CH	4.11 (m)	
23	Sucrose	C4H	3.45 (m)	all
		C2H	3.54 (m)	
		C’1H2	3.67 (m)	
		C3H	3.74 (m)	
		C’6H2_,_ C6H2	3.80 (m)	
		C’5H_,_ C5H	3.87 (m)	
		C’4H	4.03 (m)	
		C’3H	4.19 (d)	
		C1H	5.39 (d)	
24	Thymidine *	C1’H ribose	6.28 (t)	NEC2
		C6H ring	7.63 (d)	
25	Tyrosine ^2^	C3H, C5H ring	6.88 (d)	all
		C2H, C6H ring	7.17 (d)	
26	Uridine	C1’H ribose, C5H ring	5.89 (dd)	NEC2, EC1, EC2
		C6H ring	7.85 (d)	
27	Valine	*γ*CH_3_	0.99 (d)	all
		*γ*’CH_3_	1.04 (d)	

^1^ All metabolites were identified as putatively annotated compounds (level 2) according to the MSI guidelines unless stated otherwise; ^2^ putatively identified as a member of a compound class based on spectral similarity to known compounds of a chemical class (level 3); * not included in chemometric analysis.

**Table 2 plants-12-02869-t002:** Differentially expressed metabolites (VIP > 1) and performance of PLA-DA models. The values of Q2 and R2 estimate predictive ability and the goodness of fit, while the component number represents the optimized number of latent variables.

Model	Q2	R2	Component Number	Important Features
General (all *calli*)	0.990	0.995	4	Fructose Sucrose Glucose
EC1 vs. EC2	0.999	0.999	2	Fructose Pro Sucrose Choline
NEC1 vs. NEC2	0.996	0.998	1	Sucrose Choline Fructose Pro GABA
EC1-2 vs. LTC	0.945	0.999	6	Choline Lactate Ala Sucrose Gln Fructose
NEC1-3 vs. LTC	0.997	0.999	5	Fructose Glucose Sucrose Choline

**Table 3 plants-12-02869-t003:** List of differentially expressed metabolites in the volcano plot analysis. F, the fold change, is the ratio of the average concentration of the metabolite in the test *callus* to the average of the LTC.

EC1			EC2		
Metabolite	log_2_(F)	−log_10_(*p*)	Metabolite	log_2_(F)	−log_10_(*p*)
Lactate	3.0157	8.9975	Lactate	3.0793	8.8111
Glucose	1.9727	4.5264	Pro	2.9962	8.1141
Ala	1.9164	5.2626	Glucose	2.854	6.7433
GABA	1.8693	8.539	Fructose	2.6606	8.3922
Fructose	1.3182	4.4695	β-Ala	2.3036	8.811
Uridine	1.2561	2.4695	Uridine	2.2245	5.4269
Pro	1.1759	2.5534	GABA	2.181	8.1141
Asn	−1.1302	1.853	Ala	1.9012	6.6857
			Adenosine	1.8386	2.5225
			Choline	−1.0833	9.1291
			D-lactate	−1.5989	7.4762
			Chlorogenate	−2.0409	1.3749
			Asn	−2.262	2.9808
			Tyr	−2.8503	1.4842
**NEC1**			**NEC2**		
Metabolite	log_2_(F)	−log_10_(*p*)	Metabolite	log_2_(F)	-log_10_(*p*)
GABA	4.0647	8.1504	Uridine	4.8861	3.8556
Glucose	3.3213	7.6686	Adenosine	4.3811	4.0685
Fructose	2.8625	7.9148	Fructose	3.4559	7.8242
Pro	2.2208	5.8718	Pro	3.3086	3.0872
β-ala	1.4617	5.9125	Glucose	2.7991	4.6198
Ile	−1.0624	1.4783	β-Ala	2.5844	3.4663
Ethanolamine	−1.0945	4.0848	GABA	2.4035	5.9496
Gln	−1.8525	1.6637	Lactate	2.3676	3.7499
Choline	−2.3282	10.34	Ala	2.114	3.8435
Chlorogenate	−2.6901	3.8669	Formate	1.8583	2.1599
D-lactate	−4.9126	8.3448	Val	1.6334	1.5457
			Acetate	1.4829	3.24
			Ile	1.0863	1.0379
			D-lactate	−1.1509	4.6059
			Sucrose	−3.1713	5.4015
**NEC3**					
Metabolite	log_2_(F)	−log_10_(*p*)			
Glucose	5.3571	10.719			
Fructose	4.3449	9.8747			
GABA	1.9708	6.9013			
Pro	1.508	4.0053			
Ile		1.3825			
Sucrose	−1.9774	4.164			
Choline	−2.3462	9.1183			
Asn	−2.4002	2.8545			
Chlorogenate	−4.3551	1.3008			
D-lactate		7.8344			

## Data Availability

Not applicable.

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
