# Peer review of "Primary Metabolite Screening Shows Significant Differences between Embryogenic and Non-Embryogenic Callus of Tamarillo (*Solanum betaceum* Cav.)"

_plants, 2023, doi:10.3390/plants12152869_

Round 1
Reviewer 1 Report
The ms deals with A NMR-based study of primary metabolites in an indirect somatic embryogenesis system of Tamarillo (Solanum betaceum Cav. A protocol of indirect somatic embryogenesis was applied to obtain embryogenic and non-embryogenic callus from leaf segments to analyse and compare the primary metabolome by Nuclear Magnetic Resonance Spectroscopy in these distinct calli possessing the same genetic back-ground callus.
The work is very interesting and well addressed. Only some minor corrections should be done before being published:
1. The quality of the pictures in Fig. 1 must be improved to appreciate the calli.
2. Y axes in Fig.7 are not readible after printing.
Line 298. ...the sugars in the can...Please, correct the sentence.
Line 530. ...described previously described....Please, avoid repetition so closely
Line 542. In all samples a fresh amunt of samples...Please, avoid to repeat samples
Line 545. Please revise the headline 4.4. Nuclear magnetic resonance and chemeometric analyses.
Author Response
Reviewer 1
- The quality of the pictures in Fig. 1 must be improved to appreciate the calli.
The calli images have been enlarged and quality enhanced. The entire figure was slightly modified.
- Y axes in Fig.7 are not readable after printing.
The Graphpad layout was converted to TIF before pasting in the manuscript to avoid printing errors.
- Line 298. ...the sugars in the can...Please, correct the sentence.
The sentence was rephrased as follows: “Consequently, the glucose and fructose found in the calli can be inferred to have derived from sucrose”.
- Line 530. ...described previously described....Please, avoid repetition so closely
“described” has been deleted.
- Line 542. In all samples a fresh amunt of samples...Please, avoid to repeat samples.
The sentence was rewritten: “In all samples, a fresh amount of callus (75 mg)…”. Furthermore, References to samples made in triplicate was also added.
- Line 545. Please revise the headline 4.4. Nuclear magnetic resonance and chemeometric analyses.
Changed accordingly.
Reviewer 2 Report
The manuscript is devoted to the study of the primary metabolites in an indirect somatic embryogenesis system of tamarillo. While the study in itself may be of general interest to the readers in the field, a number of major issues need to be addressed before further consideration is given to the manuscript.
- The title should be changed to be more clear.
- I recommend rewriting the abstract to indicate the major conclusions of the work.
- Please, clarify the choiceof only one quantitative method (NMR)?
- Please, clarify the new finding of article.
- While the conclusions drawn from these studies are intriguing, the evidence provided is suggestive rather than conclusive. To effectively support their conclusions, it is crucial to employ alternative methods and carefully select appropriate negative controls. I suggest authors write a more clearly section "Conclusions", especially biological application this data.
Minor editing of English language required
Author Response
Reviewer 2
- The title should be changed to be more clear.
The title was changed as follows: Primary metabolite screening shows significant differences between embryogenic and non-embryogenic callus of tamarillo (Solanum betaceum Cav.). Additionally, the key words were altered to reflect the change in the title: embryogenic callus and non-embryogenic callus were removed; Indirect embryogenesis; Nuclear Magnetic Resonance spectroscopy were added.
- I recommend rewriting the abstract to indicate the major conclusions of the work.
The abstract has been rewritten to focus more on the main results found.
- Please, clarify the choice of only one quantitative method(NMR)?
The objective of this work was the direct comparison of the primary metabolites in two different types of calli (embryogenic and non-embryogenic), which possess the same genotype and are under the same external conditions, since the culture media are equal. These phenotypes vary in several aspects, the most important of which is the embryogenic competence, that is, the ability to originate somatic embryos upon removal of external auxin stimulation. Therefore, a screening method that allows the quantitative analysis of a wide range of metabolites can be applied to identify putative molecular markers that can distinguish between the embryogenic and non-embryogenic cell lines. Indeed, cell lines of other species have been studied by other authors using NMR as the unique approach used since it is specific enough to analyse the metabolites (see the quoted literature). The results we have obtained show that the identification of metabolites by NMR is a simple and effective process to detect cells involved in an embryogenic pathway, which can be used to detect these cells early and use primary metabolites as biochemical markers of embryogenesis, not only in tamarillo but also in other species. Moreover, NMR is a highly reproducible methodology that has been applied in similar cell systems and the amount of data generated are wide enough for robust statistical analysis through the methods we have employed.
Although it can be argued NMR suffers from poor resolution and sensitivity when compared to chromatography-based metabolomics, the ease of sample preparation and reproducibility of results as well as relative simplicity of metabolite identification, make it interesting technique for metabolomic applications and NMR-based metabolomics is widely used in studying plant metabolism.
The introduction has been changed to reflect this comment (lines 104-106).
- Please, clarify the new finding of article.
The investigation brings new data regarding differences between the metabolic profiles of embryogenic and non-embryogenic calli. Since indirect embryogenesis is the most common type of embryogenic induction another species, in particular in trees, our approach can be used to identify putative biochemical markers of embryogenesis in other model systems, in particular tree species. In general, carbohydrate metabolism, mainly sugars derived from sucrose, that is fructose and glucose, are the principal metabolic parameters that predict the differences of the studied phenotypes (namely embryogenic and non-embryogenic).
These findings are reflected in the abstract and in section 5 (conclusions and future perspectives).
- While the conclusions drawn from these studies are intriguing, the evidence provided is suggestive rather than conclusive. To effectively support their conclusions, it is crucial to employ alternative methods and carefully select appropriate negative controls. I suggest authors write a more clearly section "Conclusions", especially biological application this data.
Please, see comment 3. We believe it answers the first part of your comment.
In what concerns the use of negative controls, we see this suggestion as very difficult to manage in our system, because our main goal was only to compare the metabolic profile of both type of calli trying to identify markers of the embryogenic and the non-embryogenic condition. What would be negative controls? Embryogenic and non-embryogenic calli cultured on media without auxin? This approach could be applied. However, it would introduce more variables, since under these conditions the non-embryogenic calli enter senescence and the embryogenic calli begin to form embryos, which would be reflected in new metabolic conditions that would complicate the analysis we carried out.
A final paragraph (lines 512-535) has been added to the discussion to better frame this matter.
Round 2
Reviewer 2 Report
The authors have satisfactorily addressed most of my concerns. This revision has also significantly improved the manuscript.
Minor editing of English language required